# Effect of Packaging Type and Aging on the Meat Quality Characteristics of Water Buffalo Bulls

**DOI:** 10.3390/ani12020130

**Published:** 2022-01-06

**Authors:** Muhammad Hayat Jaspal, Iftikhar Hussain Badar, Muhammad Usman Ghani, Muawuz Ijaz, Muhammad Kashif Yar, Adeel Manzoor, Jamal Nasir, Kashif Nauman, Muhammad Junaid Akhtar, Abdur Rahman, Faisal Hussnain, Arfan Ahmad

**Affiliations:** 1Department of Meat Science and Technology, Faculty of Animal Production and Technology, University of Veterinary and Animal Sciences, Lahore 54000, Pakistan; hamilton.usman@gmail.com (M.U.G.); adeel.rehman@uvas.edu.pk (A.M.); jamalnasir@uvas.edu.pk (J.N.); drkashif@uvas.edu.pk (K.N.); raojunaid452@gmail.com (M.J.A.); 2College of Food Science, Northeast Agricultural University, Harbin 150030, China; 3Department of Animal Sciences, CVAS-Jhang 35200, University of Veterinary and Animal Sciences, Lahore 54000, Pakistan; muawuz.ijaz@uvas.edu.pk (M.I.); Kashif.yar@uvas.edu.pk (M.K.Y.); abdurrehman@uvas.edu.pk (A.R.); 4Department of Poultry Production, Faculty of Animal Production and Technology, University of Veterinary and Animal Sciences, Lahore 54000, Pakistan; faisal.hussnain@uvas.edu.pk; 5Institute of Microbiology, University of Veterinary and Animal Sciences, Lahore 54000, Pakistan; iffivet@uvas.edu.pk

**Keywords:** buffalo beef, meat packaging technologies, meat quality characteristics, meat-eating quality

## Abstract

**Simple Summary:**

The water buffalo is found worldwide, but mainly in Asian countries, i.e., India, Pakistan, and China. Buffalo meat can be a viable option to fulfill the future protein demands of the world’s population. Presently, very little information is available regarding buffalo meat quality attributes under different packaging types. Therefore, this study was designed to evaluate the effect of packaging type and aging time on the meat quality characteristics (instrumental color, WBSF, cooking loss, thiobarbituric acid reactive substances (TBARS), and total volatile basic nitrogen (TVB-N) of *longissimus lumborum* steaks. The results showed that vacuum packaging and aging were the most effective in decreasing the WBSF values of buffalo meat.

**Abstract:**

The present study determined the effect of the packaging type and aging time on the meat quality of water buffalo (*Bubalus bubalis*) bulls. A total of *n* = 36 *longissimus lumborum* (LL) muscles from *n* = 18 buffalo bulls were obtained. Half LL muscles were packed in modified atmosphere packaging (Hi-O_2_ MAP), vacuum packaging (VP), and oxygen-permeable packaging (OP) on day 1, while the other half were aged for 7 days. Meat instrumental color, cooking loss, Warner–Bratzler shear force (WBSF), thiobarbituric acid reactive substances (TBARS), and total volatile basic nitrogen (TVB-N) of the LL steaks were analyzed, both on unaged and aged buffalo meat. Color CIE L* and C* values on all display days and a* on the first 4 days of the simulated retail display under Hi-O_2_ MAP packaging were significantly higher than those of the VP and OP. WBSF and TBARS values were also higher under Hi-O_2_ MAP as compared to the other packaging. Steaks under OP exhibited lower cooking loss but higher TVB-N values than the MAP and VP. The 7-day-aged buffalo meat indicated higher instrumental color (L*, a* and C*), cooking loss, and lower WBSF values than fresh meat. This study concluded that Hi-O_2_ MAP improved the color; however, it negatively influenced the buffalo meat’s WBSF and TBAR values. Furthermore, VP and aging were the most effective in decreasing the WBSF values of buffalo meat.

## 1. Introduction

Retail packaging has prime importance in the meat industry, as it serves many purposes, including product identification, promotion, and protection [1]. Meat packaging is also used to delay microbial spoilage, reduce weight loss, maintain the color and permit some intrinsic enzymes to improve the tenderness [2,3,4]. Traditionally, different types of retail packaging such as oxygen-permeable (OP) packaging, high oxygen modified atmosphere packaging (Hi-O_2_ MAP), and vacuum packaging (VP) are used in the meat industry. Hi-O_2_ MAP (80% O_2_:20% CO_2_) is commonly used for retail display as it enhances the red color and prolongs the shelf-life of meat, compared to OP packaging [5]. High oxygen in the MAP maintains the myoglobin in oxygenated form [6], while carbon dioxide retards the growth of aerobic bacteria [7]. However, Hi-O_2_-MAP has been linked with reduced tenderness and negative flavor due to protein and lipid oxidation [8,9]. At the same time, VP has been famous for improving meat tenderness and extending shelf life [10]. Since removing air eliminates the oxidizing effect in VP, it allows longer aging times to delay lipid oxidation and meat discoloration [11,12,13,14]. Other considerable advantages of VP are that it requires less space, is easy to handle, and provides a longer shelf-life [15].

Beef aging is a technique widely used in the meat industry to enhance tenderness and produce a uniform product that is highly acceptable to the consumers, and therefore, extensive work has been done in this area [4,16,17,18,19,20]. Almost eight out of ten consumer decisions to repurchase meat are based on the tenderness of the meat, and for that reason, its better understanding is a significant challenge, mainly when the rate of aging varies among different species and different types of muscles [21]. According to McKenna et al. [22], color during retail display is the consumers’ main quality criteria and is associated with freshness and wholesomeness when purchasing meat and meat products. Alongside improving meat tenderness, aging also improves the color characteristics of the meat [23].

Different packaging types have been widely used to improve the retail shelf life of cattle meat [23,24,25,26,27,28]. However, the effect of the packaging type on the quality characteristics of buffalo meat has not been extensively studied. The water buffalo (*Bubalus bubalis*) inhabits many countries worldwide; 97.2% of the total buffalo population is found in Asia [29], particularly in India, Pakistan, and China, while 1.9% in Africa and the rest is present in Europe and Latin America [30,31]. Therefore, the principal objective of this study was to investigate the effect of Hi-O_2_ MAP, VP, and OP packaging on color, tenderness, and cooking loss of fresh and aged meat of young buffalo bulls.

## 2. Materials and Methods

### 2.1. Animal Rearing

Buffalo bulls (*n* = 18, Nili-Ravi breed) were reared at a commercial fattening farm Big Feed (Pvt.) Ltd., Lahore, Pakistan. Animals were raised under a similar feeding strategy (feed-lot fattening) and had the same pre-slaughter management conditions throughout the rearing period. At 24 months of age (average live weight 290 ± 10 kg), the animals were transported to the lairage facility of the University of Veterinary and Animal Sciences, Lahore, Pakistan.

### 2.2. Animal Slaughtering

Before slaughter, all animals were kept off-feed for 12 h with free access to water for the hygienic dressing of the carcasses. All animals were slaughtered by the Halal Slaughtering method described in Pakistan Halal Standards, PS 3733:2013 at the commercial slaughterhouse of Department of Meat Science and Technology, University of Veterinary and Animal Sciences, Lahore, Pakistan. After evisceration, carcasses were divided into two equal sagittal halves, hung from the Achilles tendon, and shifted to the walk-in chiller. The carcasses were chilled at 0–2 °C for 24 h.

### 2.3. Muscle Sampling and Packaging

After chilling, *longissimus lumborum* (LL) muscles from the 13th thoracic vertebrae to the lumbosacral junction were excised from both sides of each carcass. The left-side LL was vacuum packed using a C300 twin-chamber vacuum packing machine (Multivac Ltd., Wolfertschwenden, Germany) and placed in a dark room at 0–1 °C for 7 days aging period. The right-sided LL was cut into 9 steaks, each having a thickness of 2.5 cm [4]. A similar procedure was adopted for the left-sided LL after 7 days of aging. A brief layout of the experimental design is shown in Table 1. Three steaks were randomly assigned the following packaging:High oxygen (Hi-O_2_) MAP (80% O_2_:20% CO_2_) using T200 MAP machine (Multivac Ltd., Swindon, UK)Vacuum packaging (VP) using C300 twin-chamber vacuum packing machine (Multivac Ltd., Wolfertschwenden, Germany)Overwrapped with oxygen permeable (OP) film in food-grade polystyrene trays

All packaged steaks underwent simulated display in a horizontal display chiller (Model: S80100VVC, Tecnodom SPA, Vigodarzere, Italy), working at 0–4 °C for 7 days to check the color stability. The light source was provided by fluorescent tubes (TLD 36 W/33–640, Philips, Karachi, Pakistan) with 500 lux light intensity recorded with a digital lux meter (AR823, Bestone Industrial Ltd., Shenzhen, China) for 12 h each day. The same procedure was adopted for 7-day-aged samples. The MAP film (PET-PVDC-PE) with a permeability of 5 cm^3^/24 h/m^2^/atm oxygen, 20 cm^3^/24 h/m^2^/atm carbon dioxide, and 4 g/24 h/m^2^ of water vapor was used, while plastic trays were made of polypropylene. Vacuum bags were made of polyethylene (PA/PE 90) with absorbency: 2.6 g/m^2^.d of water vapor, 50 cm^3^/m^2^ of oxygen, 150 cm^3^/m^2^ of carbon dioxide, 10 cm^3^/m^2^ of nitrogen.

### 2.4. Meat Quality Analysis

#### 2.4.1. Instrumental Color

Samples packed in MAP, VP and OP were evaluated for color parameters for seven consecutive days. Color parameters: lightness (L*), redness (a*), and chroma (C*) of the steaks were measured at three different sites of each steak using chroma meter (Konica Minolta CR-410, Tokyo, Japan). Then these values were averaged for statistical analysis. Minolta Chroma meter was calibrated each time using standard white tile (L* = 94.93, a* = −0.13, and C* = 2.55), CR-A44 No.15533024 provided by the manufacturer. The colorimeter used D-65 illuminant, 50 mm aperture, and 2° observer.

#### 2.4.2. Cooking Loss

After 7 days of simulated retail display, steaks were weighed and vacuum packed in polyethylene bags (150 × 200, PA/PE 90), using C300 twin-chamber vacuum packing machine (Multivac Ltd., Wolfertschwenden, Germany). The samples were cooked in a water bath (WNB45, Memmert GmbH + Co. KG, Schwabach, Germany) operating at 82 °C [32]. The core temperature of steaks was recorded by a digital thermometer (TP300, E-Maker Technology Co., Ltd., Shenzhen, China). After attaining the core temperature of 72 °C, steaks were removed from the water bath and weighed again. The weight of steaks before and after cooking was recorded using a digital weighing balance (SF-400, Zhejiang Tiansheng Electronic Ltd., Zhejiang, China). The cooking loss was measured as a percentage, as the ratio of the difference between the weights before and after cooking to the weight of steaks before cooking [33].

#### 2.4.3. Warner-Bratzler Shear Force

After cooking, steaks were placed in polystyrene trays and cooled down overnight at 0–2 °C [34]. The next day, 1 cm^3^ cross-section meat strips were obtained using scalpel-handle blades with a fiber direction parallel to the strip length [35]. Warner–Bratzler shear force values were measured in newton per centimeter square (N/cm^2^) by TA.XTplus texture analyzer (Stable Micro System Ltd., Godalming, UK) using a ‘V-Slot’ blade. Minimum of three shear force values were recorded for each sample and averaged for statistical analysis.

#### 2.4.4. Lipid Oxidation

Lipid oxidation was determined through thiobarbituric acid reactive substances (TBARS) values following the method described by Zhang et al. [36] with some modifications. After 7 days of simulated retail display, 5 g of the meat samples were taken from three different sites of each steak’s surface. The sample was transferred to a 50 milliliter (mL) conical tube, and 1 mL of 10% BHA (*Butylated hydroxyanisole*) solution (prepared in ethanol) was added along with 14 mL double-distilled water and homogenized using a T 25 digital Ultra-Turrax homogenizer (IKA-Werke GmbH & Co., Staufen, Germany) at 11,000 rpm for 15 s. After that, homogenate (1 mL) was added to a test tube along with a 2 mL solution of TBA/TCA (thiobarbituric acid/trichloroacetic acid), vortexed for mixing, filtered, and was incubated in a water bath operating at 80 °C for 90 min; then, the sample was allowed to cool down, vortexed again, and centrifuged for 15 min at 2000 rpm. The absorbance of the supernatant was measured at 532 nm in a spectrophotometer (Shimadzu UV-1800 spectrophotometer, Kyoto, Japan). The TBARS values were expressed as mg of malonaldehyde (MDA) per kg of meat samples.

#### 2.4.5. Total Volatile Basic Nitrogen (TVB-N)

The TVB-N content was calculated as described by Chen et al. [37] with some modifications. The 10 g representative sample was collected from three different sites of each steak’s surface. The samples were homogenized in 100 mL distilled water at 11,000 rpm using a T 25 digital Ultra-Turrax homogenizer (IKA-Werke GmbH & Co., Staufen, Germany). Then the homogenate was shifted to the distillation tube and 1 g magnesium oxide (MgO) was added. The distillation tube was connected to the Kjeldhal Apparatus K 355 (BÜCHI Labortechnik AG, Flawil, Switzerland). The volatile basic nitrogen was absorbed using a 20 mL boric acid solution (2%) containing color indicators (i.e., 0.1 g methylene blue and 0.1 g methyl red). After, the titration was performed using 0.01 N hydrochloric acid (HCl) solution. The TVB-N values were expressed in mg/100 g and calculated as follows:TVB-N = (Vol. of HCL used × Normality of acid × 14)/Sample weight (g) × 100

### 2.5. Statistical Analysis

Statistical analysis was performed by SAS software (Version 9.1.3; 2002–2004, Cary, NC, USA). Effects of Hi-O_2_ MAP, VP, OP, and aging time on meat quality characteristics were analyzed through factorial ANOVA, using PROC GLM (general linear model), including packaging type and aging time as a fixed effect, and their interaction was also tested. There were non-significant interactions between packaging type and aging time. Therefore, interactions effects are not shown in the tables. Significant treatment means were compared using the Duncan multiple range DMR test [38]. The significance level was considered at *p* ≤ 0.05.

Following statistical model was used:Y = μ + F1i + F2j + (F1 F2) ij + εij
where Y = response variable, μ = overall population mean, F1i = fixed effect of packaging type, F2j = fixed effect of aging time, (F1 F2) ij = interaction effect of packaging type and aging time, and εij = random error.

## 3. Results and Discussion

### 3.1. Instrumental Meat Color

The influence of packaging type on instrumental color is presented in Table 2. Packaging type significantly affected L*, a*, and C* values throughout the 7 days of simulated retail display. The L* value of buffalo meat packed in Hi-O_2_ MAP was highest (*p* ≤ 0.05), followed by OP and VP, respectively. A decreasing trend of L* value was found in all three packaging types throughout the 7 days of simulated retail display. The a* value of the buffalo LL steaks packed in Hi-O_2_ MAP was highest (*p* ≤ 0.05) up to day 6, whereas the a* value of OP packed buffalo meat was higher than VP on days 1 and 2. It was similar to the a* value of VP buffalo meat on day 3, while it was lower on days 4, 5, 6, and 7 than VP meat. The a* value was reduced throughout the 7 days of retail display in all three types of packaging. The C* value of buffalo meat packed in Hi-O_2_ MAP was significantly higher than OP and VP meat throughout the 7 days of retail display. OP exhibited a higher C* value than VP from day 1 to day 4, while on day 5 and day 6, VP indicated significantly higher C* values than OP buffalo meat. The C* value decreased with time in all three types of packaging.

Sekar et al. [39] researched buffalo meat and compared its color in aerobic, vacuum, and MAP packaging. In the study, the meat color was judged by a trained panel, and the scientists found that MAP gives a bright red color to the meat, followed by aerobic packaging and vacuum packaging, respectively. Moczkowska et al. [27] studied instrumental color in MAP and vacuum skin packaging (VSP) comparing with control and observed similar results of MAP meat being most reddish (higher a*) and VSP meat least reddish. Lagerstedt et al. [25] and Vitale et al. [23] agree that Hi-O_2_ MAP enhances meat color parameters. The bright red color in Hi-O_2_ MAP was due to higher oxygen concentration in Hi-O_2_ MAP, which allows the formation of a thick oxymyoglobin layer due to the deeper penetration of oxygen [40]. At the same time, the possible reason for an increase in lightness might be the oxidizing conditions in Hi-O_2_ MAP that cause changes in the meat protein structure, leading to greater light dispersion hence higher L* value [41]. Since air is removed in VP, more myoglobin is in deoxy form [42]. Therefore, the a* value of VP meat was less than in Hi-O_2_ MAP and OP meat.

As shown in Table 2, aging significantly influenced the instrumental color. The L* value of 7-day-aged buffalo meat was slightly higher for the first 5 days than fresh buffalo meat. Significant results were observed for the L* value on days 3 and 5 only. On days 6 and 7, aged buffalo meat indicated a lower L* value. The 7-day-aged buffalo meat had a higher L* value on days 1 to 4, with a lower a* on the subsequent 3 days.

Similarly, significant results were observed on day 1. The C* value was also significantly higher in aged buffalo meat on days 1 and 2. These findings agree with the observations of Vitale et al. [23], who showed that lightness, redness, and chroma values of *longissimus thoracis* et lumborum steaks aged 3 to 21 days under VP conditions were higher than unaged meat (48 h post mortem). While studying the influence of different aging times in VP on the instrumental color of young bulls’ *longissimus dorsi* steaks, Lagerstedt et al. [25] also found increased lightness, redness, and yellowness in 5- and 15-day-aged meat as compared to unaged meat (72 h post mortem). The differences between the lightness, redness, and chroma values of pre-aged and aged meat can be justified by VP’s blooming ability. As Ledward [43] described, the blooming of meat relies on several factors, including oxygen concentration, its diffusion into the meat, and muscle oxygen consumption rate. O’keeffe and Hood [44] and Bendall and Taylor [45] explained that the difference in color parameters of fresh and aged meat could reduce oxygen consumption rate over time in a vacuum pack. Therefore, oxygen penetrates deeper due to limited enzymatic activity, and a thick layer of oxymyoglobin is formed [46]. The color stability of fresh and aged buffalo meat in the present study did not differ. This was in agreement with the results of Vitale et al. [23] and Lagerstedt et al. [25]. However, Ledward [43] observed that aged meat turns brownish earlier as compared to fresh meat. Zakrys et al. [47] reported a negative correlation of a* and positive correlation of L* with days of the display, meaning that meat became less red and lighter with the passage of days in the retail display, and the present study partially agrees with this correlation as redness decreased during the 7 days of simulated retail display, while lightness increased up to 3 days only.

### 3.2. Cooking Loss

As shown in Table 3, the buffalo meat in MAP and VP showed a significantly higher cooking loss than OP buffalo meat at the end of 7 days of simulated retail display (Table 3). The higher cooking loss in MAP could be due to the higher protein oxidation leading to limited cytoskeletal protein degradation that enhanced cell shrinkage in the overall muscle structure [48].

Cooking loss was higher in aged buffalo meat (Table 3). Wyrwisz et al. [49] also observed increased cooking loss after 7 days in vacuum packaging. This increased cooking loss might be due to the protein breakdown and loosening of muscle structure during aging [48,50,51,52].

### 3.3. Warner–Bratzler Shear Force (WBSF)

The effect of packaging type on WBSF values is presented in Table 3. WBSF values were significantly lowest for VP, followed by OP and MAP, respectively (Table 3). Lagerstedt et al. [25,35] and Vitale et al. [23] also found similar results. Zakrys et al. [47] found that when oxygen concentration is increased from 10% to 80%, the increase in concentration tends to increase shear force values during storage of 15 days. The lowest WBSF values in VP were reported because the tenderizing process by proteolytic enzymes continues in VP [23]. This is also supported by the study conducted by Sekar et al. [39], who found that buffalo meat stored in VP had a significantly longer sarcomere length and smaller myofibrillar fragmentation index than meat stored in Hi-O_2_ MAP and OP. At the same time, these two proposed theories that could explain the increase in WBSF values in Hi-O_2_ MAP. The first is the inactivation of enzymes involved in oxidation conditioning, leading to a slower tenderization process [53]. Second, protein cross-linkages are formed in a high oxygen environment, resulting in tougher meat [5,54,55]. Usually, VP is not preferred by consumers due to the purple color of the meat. However, Scandinavian customers preferred steaks from oxygen-free packaging in terms of willingness to pay more, sensory quality, and overall liking [56], which indicates better quality is preserved in VP.

As shown in Table 3, aging significantly reduced the WBSF (improved tenderness) of the buffalo meat compared to unaged buffalo meat (Table 3). Vitale et al. [23] studied the effect of 0, 3, 6, 8, 14, and 21 days aging on tenderness and found that aged meat was tenderer than unaged meat. Similarly, Moczkowska et al. [27] observed lower WBSF values in 14 and 28 days vacuum skin-pack aged meat compared to the unaged meat. Many authors have observed a similar tenderizing effect of aging in cattle beef [35,57,58,59,60]. The decrease in WBSF values could be due to the breakdown of myofibrillar cytoskeletal proteins by proteolytic enzymes, i.e., calpains, cathepsins, and caspases [48,50,51,52]. In this study, WBSF values of young buffalo meat were under the acceptable threshold of 40.2 N for consumers, as indicated by Huffman et al. [61].

### 3.4. Lipid Oxidation

Packaging type significantly affected TBAR values (Table 3). The highest TBAR values were observed in MAP, whereas the lowest TBAR values were noticed in VP at the end of the 7 days of simulated retail display. The present study results are in accordance with the findings of Clausen et al. [62]. The authors reported that packaging type significantly impacts the TBAR values, the lowest TBAR values in anaerobic packaging and the highest in aerobic packaging. Similarly, Amaral et al. [63] presented that oxygen exposure is a critical factor for lipid oxidation in meat products; higher oxygen exposure leads to enhanced lipid oxidation, following the present study’s results. In other studies, authors found similar results, lower TBAR values in vacuum-packed meat and highest lipid oxidation values in modified atmosphere packaging with high oxygen concentration [64,65].

As shown in Table 3, aging also significantly affected TBAR values. The highest TBAR values were observed in 7 days aged meat. Ismail et al. [66] found similar findings in a study on beef rounds. The authors showed that lipid oxidation increased with the increasing aging time. Likewise, Karami et al. [67] also found that TBAR values were increased with increasing aging time. In contrast, Rant et al. [68] and Marrone et al. [69] found a non-significant impact of aging time on the oxidative stability of meat.

### 3.5. Total Volatile Basic Nitrogen (TVB-N)

Packaging type significantly affected TVB-N values; the highest TVB-N values (14.30 mg/100 g) were observed in OP, followed by VP and MAP. However, the highest values were under the acceptable threshold level of TVB-N, which is less than 20 mg/100 g [70]. The TVB-N compounds are toxic nitrogenous compounds resulting from microbial contamination, and enzymatic actions can ultimately lead to sensory changes; they are an excellent indicator of meat safety and freshness [71]. Mansur et al. [72] exhibited similar results that aerobic storage showed the highest level of TVB-N during 9-day storage than the vacuum packaging. Likewise, Lyu et al. [73] found that MAP meat had lower TVB-N than vacuum packaged meat.

Aging significantly affected the TVB-N values; the highest values were noticed in 7-day-aged meat. Similarly, Chen et al. [37] showed that TVB-N level increased with increasing storage duration of meat. In a study, the authors found that the TVB-N level increased during a 46-day-storage period of beef loins [73]. In a recent study, Azarifar et al. [74] reported that the TVB-N level was increased with storage time that exceeded the unacceptable limit after 12 days at 4 °C.

## 4. Conclusions

In conclusion, Hi-O_2_ MAP improved the buffalo meat’s color and shelf life compared to the OP; however, it negatively affected the WBSF and TBARS values. While VP decreased the WBSF values and improved the color shelf life, compared to Hi-O_2_ MAP, it has a drawback of giving purple-colored meat. The aging of meat in different simulated retail displays for 7 days improved the meat color and tenderness. Therefore, different packaging types and aging times for buffalo meat should be chosen according to the customers’ requirements to ensure high-quality meat and meat products.

## Figures and Tables

**Table 1 animals-12-00130-t001:** A brief layout of experimental design.

Total Animals	Muscle	Ageing Time	Packaging
Buffalo bulls *n* = 18 24 months Carcass weight (130 ± 10 kg)	Both sided *longissimus lumborum* (LL) muscles were removed	0 day Right sided LL is divided into steaks (*n* = 9) of 2.5 cm thickness	MAP (3 steaks)
			Vacuum packed (3 steaks)
			Overwrapped (3 steaks)
		7 day Left sided LL is divided into steaks (*n* = 9) of 2.5 cm thickness after 7 days of aging	MAP (3 steaks)
			Vacuum packed (3 steaks)
			Overwrapped (3 steaks)

**Table 2 animals-12-00130-t002:** Effect of packaging and ageing on color CIE L*, a* and C* of buffalo bulls *longissimus lumborum* steaks during the 7 days of retail display.

Parameters	Day	Packaging			*p*-Value	Ageing		*p*-Value
		MAP	VP	OP		0 day	7 day	
Lightness L*	1	49.37 ± 0.23 ^a^	42.22 ± 0.18 ^c^	46.69 ± 0.15 ^b^	*	45.89 ± 0.44	46.29 ± 0.43	ns
	2	50.01 ± 0.21 ^a^	42.32 ± 0.16 ^c^	46.51 ± 0.12 ^b^	*	46.18 ± 0.44	46.39 ± 0.46	ns
	3	49.98 ± 0.19 ^a^	42.31 ± 0.16 ^c^	46.32 ± 0.18 ^b^	*	46.00 ± 0.43 ^b^	46.41 ± 0.47 ^a^	*
	4	49.86 ± 0.20 ^a^	42.17 ± 0.21 ^c^	45.73 ± 0.25 ^b^	*	45.77 ± 0.44	46.07 ± 0.50	ns
	5	49.74 ± 0.23 ^a^	42.13 ± 0.23 ^c^	45.58 ± 0.22 ^b^	*	45.51 ± 0.46 ^b^	46.12 ± 0.47 ^a^	*
	6	48.70 ± 0.23 ^a^	42.19 ± 0.22 ^c^	44.98 ± 0.17 ^b^	*	45.38 ± 0.42	45.21 ± 0.39	ns
	7	48.71 ± 0.21 ^a^	41.84 ± 0.23 ^c^	44.85 ± 0.16 ^b^	*	45.21 ± 0.43	45.06 ± 0.41	ns
Redness a*	1	17.98 ± 0.20 ^a^	15.48 ± 0.13 ^c^	17.41 ± 0.18 ^b^	*	16.52 ± 0.15 ^b^	17.39 ± 0.23 ^a^	*
	2	17.22 ± 0.15 ^a^	15.20 ± 0.10 ^c^	16.40 ± 0.19 ^b^	*	16.15 ± 0.15	16.40 ± 0.18	ns
	3	16.55 ± 0.22 ^a^	15.23 ± 0.12 ^b^	15.22 ± 0.15 ^b^	*	15.53 ± 0.13	15.80 ± 0.14	ns
	4	15.72 ± 0.13 ^a^	15.01 ± 0.09 ^b^	14.40 ± 0.15 ^c^	*	14.98 ± 0.12	15.11 ± 0.13	ns
	5	14.98 ± 0.15 ^a^	14.89 ± 0.12 ^a^	13.45 ± 0.17 ^b^	*	14.48 ± 0.15	14.40 ± 0.16	ns
	6	14.37 ± 0.21 ^a^	14.80 ± 0.15 ^a^	12.45 ± 0.20 ^b^	*	13.95 ± 0.19	13.80 ± 0.22	ns
	7	13.49 ± 0.20 ^b^	14.71 ± 0.13 ^a^	11.32 ± 0.22 ^c^	*	13.24 ± 0.23	13.11 ± 0.26	ns
Chroma C*	1	20.61 ± 0.13 ^a^	15.68 ± 0.10 ^c^	19.72 ± 0.19 ^b^	*	18.33 ± 0.29 ^b^	19.01 ± 0.34 ^a^	*
	2	19.85 ± 0.11 ^a^	15.40 ± 0.10 ^c^	18.69 ± 0.12 ^b^	*	17.83 ± 0.27 ^b^	18.14 ± 0.28 ^a^	*
	3	19.29 ± 0.10 ^a^	15.40 ± 0.11 ^c^	17.50 ± 0.16 ^b^	*	17.37 ± 0.24	17.42 ± 0.24	ns
	4	18.72 ± 0.12 ^a^	15.15 ± 0.12 ^c^	15.92 ± 0.18 ^b^	*	16.66 ± 0.24	16.53 ± 0.24	ns
	5	18.23 ± 0.14 ^a^	15.06 ± 0.14 ^b^	14.48 ± 0.17 ^c^	*	16.01 ± 0.25	15.83 ± 0.26	ns
	6	17.42 ± 0.19 ^a^	14.86 ± 0.13 ^b^	13.65 ± 0.16 ^c^	*	15.36 ± 0.24	15.26 ± 0.27	ns
	7	16.06 ± 0.26 ^a^	14.83 ± 0.12 ^b^	12.72 ± 0.14 ^c^	*	14.63 ± 0.24	14.44 ± 0.25	ns

^a,b,c^ Means within a row followed by different superscript differ significantly at *p* ≤ 0.05. MAP: modified atmosphere packaging; VP: vacuum packaging; OP: oxygen permeable. * *p* ≤ 0.05; ns means non-significant.

**Table 3 animals-12-00130-t003:** Effect of packaging and ageing on cooking loss (%) and tenderness (N/cm^2^) of buffalo bulls *longissimus lumborum* steaks after 7 days of retail display.

Parameters	Packaging			*p*-Value	Ageing		*p*-Value
	MAP	VP	OP		0 day	7 day	
Cooking loss	31.62 ± 0.34 ^a^	31.52 ± 0.53 ^a^	29.12 ± 0.40 ^b^	*	29.08 ± 0.49 ^b^	30.42 ± 0.49 ^a^	*
Tenderness	44.87 ± 1.03 ^a^	28.32 ± 0.95 ^c^	33.97 ± 0.69 ^b^	***	39.57 ± 1.10 ^a^	31.86 ± 1.04 ^b^	***
TBARS	0.32 ± 0.10 ^a^	0.20 ± 0.08 ^c^	0.26 ± 0.12 ^b^	***	0.22 ± 0.11 ^b^	0.25 ± 0.11 ^a^	***
TVB-N	10.54 ± 0.25 ^b^	9.14 ± 0.18 ^c^	14.30 ± 0.28 ^a^	***	8.26 ± 0.34 ^b^	14.96 ± 0.42 ^a^	***

^a,b,c^ Means within a row followed by different superscript differ significantly at *p* ≤ 0.05. MAP: modified atmosphere packaging; VP: vacuum packaging; OP: oxygen permeable. * *p* ≤ 0.05; *** *p* ≤ 0.001.

## Data Availability

The data are not publicly available.

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
