# Peer review of "Effect of Packaging Type and Aging on the Meat Quality Characteristics of Water Buffalo Bulls"

_animals, 2022, doi:10.3390/ani12020130_

Round 1
Reviewer 1 Report
I found the manuscript easy to read and well presented. Small mistakes were highlighted and some comments made.
The information correspond with that known for normal beef and that is why I gave the novelty Average. But being water buffalo I think it is significant knowledge to have for a novel species not well known in retail.

Author Response
Thank you for your time in reviewing this manuscript. We are thankful for your comments and appreciation. We have carefully studied the comments, modified the manuscript according to your suggestions, and marked the changes using Track Changes. They certainly allowed us to improve our paper.

Reviewer 2 Report
This work focus is on the effect of the packaging and aging on the water buffalo meat quality. Overall the paper reflects sound and well presented research. Some minors should be taken into consideration for publication.
The title must be revised. I found it somewhat confusing and no suitable.
L60. "Almost 8 in ten times" please correct this and uniform the presentation.
L78-79. Please describe the conditions
L80. weight units?
L126-127. After the 7 day period the steaks were weighed (removed form the packaging) and vacuum packed again?
L248. Was visible any major difference regarding fluid expulsion among the samples?
Table 3 data on ageing samples refer to any of the packaging tested?
Author Response
Dear Reviewer,
Thank you for your time in reviewing this manuscript. We are thankful for your comments and appreciation. We have carefully studied the comments, modified the manuscript according to your suggestions, and marked the changes using Track Changes. They certainly allowed us to improve our paper.
Best regards,

Reviewer 3 Report
Manuscript ID: animals-1408770
Effect of packaging type and aging time on the meat quality characteristics of Longissimus lumborum steaks from water buffalo bulls
I think it's a very interesting and very important topic in the meat context and buffalo meat quality nowadays as regards packaging type and aging time on the meat quality.
The manuscript evaluate effect of the packaging type and aging time on the meat 25 quality of water buffalo (Bubalus bubalis) bulls.
The topic is of interest for the academics and for the people because of the results obtained and because of the potential use of buffalo meat and its application in field. There are some studies like this in literature, but not specific in this kind of product. The research is well performed, the sampling and analysis were well done.
Statistical analysis was well performed
The conclusions are of interest
The manuscript is well written and easy to understand by readers. I believe that this manuscript does not need big changes but I think you can publish the manuscript after minor revision and an improvement of discussions.
Specific suggestions
Line 58 - 60: Beef aging is a technique widely used in the meat industry to enhance tenderness and produce a uniform product that is highly acceptable to the consumers, and therefore, extensive work has been done in this area… please cite
- Smaldone, G., Marrone, R., Vollano, L., Peruzy, M.F., Barone, C.M.A, Ambrosio, R.L., Anastasio, A, 2019. Microbiological, rheological and physical-chemical characteristics of bovine meat subjected to a prolonged ageing period. Italian Journal of Food Safety, Volume 8, Issue 3, 131-136
Line 308: impact of aging time on oxidative stability of meat.. please cite
- Marrone R., Salzano A., Di Francia A., Vollano L., Di Matteo R., Balestrieri A., Anastasio A., Barone C.M.A., 2020. Effects of feeding and maturation system on qualitative characteristics of buffalo meat (Bubalus bubalis). Animals,10, 899
Author Response

(The authors gave the same response as above.)
